# Task Regularized Hybrid Knowledge Distillation For Incremental Object Detection

## Abstract

Incremental object detection (IOD) task is trapped in well-known catastrophic forgetting. Knowledge distillation has been used to overcome this problem. Previous works mainly focus on combining different distillation methods, including feature, classification, location and relation, into a mixed scheme to solve this problem. In this paper, we find two reasons of catastrophic forgetting, knowledge fuzziness and imbalance learning. We propose a task regularized hybrid knowledge distillation method for IOD task. Our method integrates knowledge selection strategy and knowledge transfer strategy. First, we propose an image-level hybrid knowledge representation by combining instance-level hard knowledge and soft knowledge to use teacher knowledge critically. Second, we propose a task-based regularization distillation loss by taking account of loss difference between old and new tasks to make incremental learning more balance. Extensive experiments conducted on MS COCO and Pascal VOC demonstrate that our method achieves state-of-the-art performance. Remarkably, we reduce the mAP gap between incremental leaning and joint learning to 6.15% under the most difficult Five-Step scenario of MS COCO, which is superior to 19.5% of the previous best method.

## 1 Introduction

The current object detection models (Ge et al., 2021) primarily follow the overall learning paradigm, where the annotations for all categories are provided prior to the learning process. This paradigm assumes that the data distribution remains fixed or stationary (Yuan et al., 2021). However, in the real world, data is dynamic and exhibits a non-stationary distribution. When a model learns from continuously incoming data, new knowledge interferes with the previously learned knowledge, resulting in catastrophic forgetting of old knowledge (McCloskey & Cohen, 1989; Goodfellow et al., 2014). To address this issue, incremental learning has been studied in recent years and has shown advancements in image classification task. However, there has been limited research on incremental object detection (IOD) task.

Knowledge distillation has been proved to be an effective method for IOD task, in which the model trained on old classes performs as a teacher to guide the training of student model on new classes. There are four kinds of distillation schemes: feature, classification, location and relation distillation. Most previous works combine feature and classification knowledge to construct their distillation methods, while the latest work combines classification distillation and location distillation to construct a response-based distillation method. In addition, various distillation losses, based on KL diversity, cross entropy and mean square error, are proposed for knowledge transfer. In summary, the keys of knowledge distillation are what knowledge should be selected from teacher and how it is transferred to student. The former question needs Knowledge Selection Strategy (KSS), while the latter needs Knowledge Transfer Strategy (KTS).

Incremental object detection face two problems. **(1)** Teacher outputs probability distributions as logits and converts them into one-hot labels as final predictions. Logits and one-hot labels are regarded as soft knowledge and hard knowledge, respectively. Soft knowledge contains confidence relations among categories, but brings knowledge ambiguity inevitably. While, hard knowledge has completely opposite effects. Therefore, how to design KSS to keep balance between accuracy and fuzziness of knowledge is a key problem. **(2)** Incremental learning should maintain old knowledge during the learning of new knowledge to overcome catastrophic forgetting, therefore how to design

KTS to keep balance between stability of old knowledge and plasticity of new knowledge is a key problem. This paper focuses on how to design effective KSS and KTS for IOD task. We demonstrate that catastrophic forgetting can be significantly alleviated by reducing knowledge fuzziness of teacher and suppressing imbalance learning between old and new tasks.

Firstly, the max confidence value of logits is always lower than its corresponding one-hot value (equal to 1), which brings knowledge ambiguity and reduces teacher's supervise ability. This means soft knowledge is not completely reliable, which should be used critically. However, previous methods ignore this keypoint. Motivated by this insight, we propose an image-level **h**ybrid **k**nowledge **r**epresentation method, named as **HKR**, by combining instance-level soft knowledge and hard knowledge adaptively to improve the exploration of teacher knowledge. Secondly, new coming data contains massive labeled objects of new classes, while contains a few unlabeled objects of old classes, therefore student trends to be dominated by new classes and falls into catastrophic forgetting. Thus it is very important to balance the learning of old and new classes. We propose a **t**ask **r**egularized **d**istillation method, named as **TRD**, by using losses difference between old and new classes to prevent student from task over-fitting effectively. We first explore imbalance learning problem for IOD explicitly.

Our contributions can be summarized as follows: **(1)** We propose a hybrid knowledge representation strategy by combing logits and one-hot predictions to make a better trade-off and selection between soft knowledge and hard knowledge. **(2)** We propose a task regularized distillation method as an effective knowledge transfer strategy to overcome the imbalance learning between old and new tasks, which relieves catastrophic forgetting remarkably. **(3)** Extensive experiments on MS COCO, Pascal VOC and OWOD scenarios demonstrate that our method achieves state-of-the-art performance.

## 2 RELATED WORKS

**Incremental Object Detection.** There are several schemes for IOD task. Li & Hoiem (2018) first proposed a knowledge distillation scheme by applying LWF to Fast RCNN (Girshick, 2015). Zheng & Chen (2021) proposed a contrast learning scheme with a proposal contrast to eliminate the ambiguity between old and new knowledge.Joseph et al. (2021b) proposed a meta-learning scheme to share optimal information across incremental tasks. Joseph et al. (2021a) introduced the concept of open world object detection(OWOD), which integrates incremental learning and open-set learning simultaneously. In addition, Li et al. (2021) first studied few-shot IOD. Li et al. (2019) designed a IOD system on edge devices. Wang et al. (2021a) presented an online incremental object detection dataset. Recently, Wang et al. (2022) proposed a data compression strategy to improve sample replay scheme of IOD. Yang et al. (2022) proposed a prototypical correlation guiding mechanism to overcome knowledge forgetting. Cermelli et al. (2022) proposed to model the missing annotations.

**Knowledge Distillation for Incremental Object Detection.** Knowledge distillation (Hinton et al., 2015) is an effective way to transfer knowledge between models with KL diversity, cross entropy or mean square error as the distillation loss. There are mainly four kinds of knowledge distillation used in IOD task: feature, classification, location and relation distillation. LwF was the first to apply knowledge distillation to Fast RCNN detector (Li & Hoiem, 2018). RILOD designed feature, classification and location distillation for RetinaNet detector on edge devices (Li et al., 2019). SID combined feature and relation distillation for anchor-free detectors (Peng et al., 2021). Yang et al. (2021a) proposed a feature and classification distillation by treating channel and spatial feature differently. ERD is the latest state-of-the-art method, combining classification and location distillation (Feng et al., 2022). Most of existing methods combine feature, classification and location distillation in composite and complex schemes to realize knowledge selection and transfer.

## 3 OUR METHOD

### 3.1 OVERALL ARCHITECTURE

We build our incremental detector on the top of YOLOX (Ge et al., 2021), a typical one-stage anchor-free detector, which can contribute to the typical verification of our method. Its overall architecture is shown in Fig4. YOLOX designs two independent branches as its classification and location heads. Firstly, hybrid knowledge representation (**HKR**) module works after the classifi-

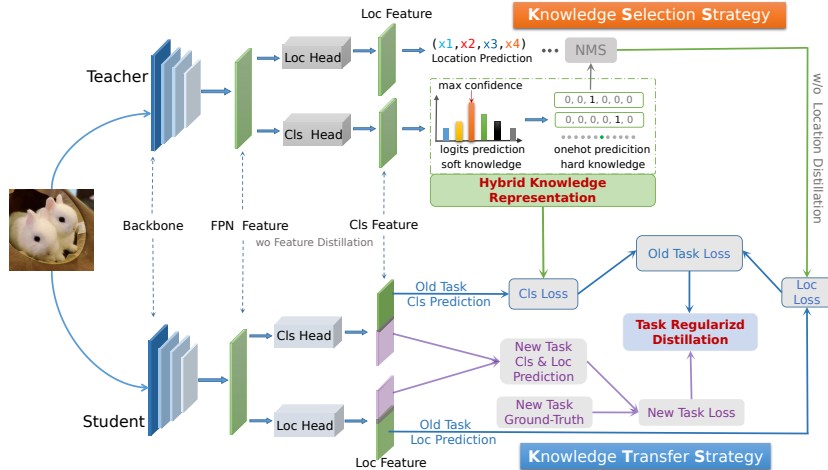

Figure 1: The overall architecture of our incremental detector ilYOLOX. Cls and Loc refer to classification and location respectively. Hybrid Knowledge Representation (HKR, Eq.3) and Task Regularized Distillation (TRD, Eq.10) refer to our proposed two components, which play roles of knowledge selection and knowledge transfer strategies, respectively.

cation head of teacher to discover valuable predictions for old classes. Secondly, task regularized distillation (**TRD**) module works between the heads of teacher and student to transfer knowledge.

## 3.2 HYBRID KNOWLEDGE REPRESENTATION

Teacher outputs probability distribution as logits and converts them to one-hot labels as final predictions. Logits and one-hot labels are regarded as soft knowledge and hard knowledge, respectively. Hinton et al. (2015) shows that soft knowledge is better than hard knowledge for classification distillation. However, although soft knowledge reflects more between-class information than hard knowledge, it also brings fuzziness to knowledge inevitably, which makes student confused during distillation learning. Actually, teacher confidence reflects knowledge quality. If teacher has high confidence about its predictions, we should further strengthen this trend so that student can feel the certainty of this knowledge. Conversely, if teacher has low confidence, we should not do that.

Therefore, the key problem is how to evaluate the quality of soft knowledge from teacher. Here, we propose to evaluate soft knowledge according to the confidence difference between the maximum value and the secondary maximum value of teacher logits. This confidence difference reflects detector's preference for Top-2 predictions. The more the detector leans towards Top-1, the higher the logit quality is. Given a batch of images, teacher outputs a batch of logits for potential objects about old categories. For each logit, if the confidence difference between its maximum confidence and secondary maximum confidence is larger than a threshold, the knowledge quality of this logit will be regarded as high, otherwise as vanilla. High quality knowledge will be represented as one-hot prediction, while vanilla knowledge will be represented as soft prediction. We compute the mean value of the confidence differences across the entire batch as the threshold to judge knowledge quality adaptively. We formulate the description above as follows:

$$ConfDiff = Conf_{first\_max} - Conf_{second\_max} \tag{1}$$

$$quality = ConfDiff > \frac{1}{N}\sum_{i}^{N} ConfDiff_i \tag{2}$$

$$Hybrid = quality \cdot Onehot + (1 - quality) \cdot Soft \tag{3}$$

where, $Conf^{N \times C}$ refers to a batch of logits with batch size of $N$ and categories of $C$. $ConfDiff^{N \times 1}$ refers to the confidence difference of each logit between its maximum confidence and secondary maximum confidence. $N$ and $i$ refers to the total number of logits and the $i^{th}$ logit. $\frac{1}{N}\sum_{i}^{N} ConfDiff_i$ is the threshold to judge knowledge quality. $quality$ defined in Eq.2 is a Boolean vector to indicate

knowledge quality for all logits. Then, $Hybrid$ predictions can be computed in Eq.3 by combining $Onehot$ predictions and $Soft$ predictions. Our method combines soft knowledge and hard knowledge dynamically to form a hybrid knowledge representation for each input image.

### 3.3 TASK REGULARIZED DISTILLATION

The learning loss of student in IOD task can be defined as following equation Eq.6. New task loss ($Loss_{new}$, Eq.5) refers to the loss supervised by the ground-truth of new classes. Old task loss ($Loss_{old}$, Eq.4) refers to the loss supervised by one-hot or soft targets from teacher. The $Loss_{cls}$ and $Loss_{loc}$ are the same as the official YOLOX, which are cross entropy loss and IoU loss respectively with coefficients of $\alpha = 1$ and $\beta = 5$. The task balance factor $\gamma$ is set to be 1 by default.

$$Loss_{old} = \alpha \cdot Loss_{cls} + \beta \cdot Loss_{loc} \tag{4}$$

$$Loss_{new} = \alpha \cdot Loss_{cls} + \beta \cdot Loss_{loc} \tag{5}$$

$$Loss_{total} = Loss_{new} + \gamma \cdot Loss_{old} \tag{6}$$

Incremental learning is easily affected by data proportion of old and new tasks. If the data proportion of new task is too large, student will be dominated by new task loss and forget old knowledge. Conversely, student will obtain much more stability to old knowledge and lack of plasticity to accept new knowledge. Therefore, the key problem of distillation learning is to keep balance between old and new tasks. Motivated by this insight, we propose a **t**ask **r**egularized **d**istillation method (TRD) to solve the imbalance learning problem. TRD method consists of two parts: task equal loss and task difference loss, which are formulated as follows:

$$Loss_{old}^{*} = \left[\frac{2 \cdot Loss_{new}}{Loss_{old} + Loss_{new}}\right] \cdot Loss_{old} \tag{7}$$

$$Loss_{new}^{*} = \left[\frac{2 \cdot Loss_{old}}{Loss_{old} + Loss_{new}}\right] \cdot Loss_{new} \tag{8}$$

$$Loss_{diff}^{*} = (Loss_{old} - Loss_{new})^2 \tag{9}$$

$$Loss_{total}^{*} = Loss_{new}^{*} + Loss_{old}^{*} + \eta \cdot Loss_{diff}^{*} \tag{10}$$

Where, $Loss_{old}$ and $Loss_{new}$ are defined in Eq.4 and Eq.5. $Loss_{old}^{*}$ and $Loss_{new}^{*}$ are the newly defined losses for old and new tasks. [] refers to the detach operation of PyTorch, which can separate a variable from graph to remove the gradient back-propagation of that part. [] operation adds two task-based balance factors to $Loss_{old}$ and $Loss_{new}$, so that $Loss_{old}^{*}$ and $Loss_{new}^{*}$ will be always equal to each other during the entire incremental learning. Therefore, we can ensure a completely dynamic balance between old and new tasks regardless of their data imbalance. $Loss_{diff}^{*}$ measures the loss difference between old and new tasks, which can further contribute to their balance learning. $\eta$ is a weighting factor. $Loss_{total}^{*}$ is the final formulation of TRD method. Compared with Eq.6, TRD emphasizes task balance explicitly by introducing two task-based balance factors (seen in Eq.7 and Eq.8) and a task-based penalty item (Eq.9), thus can prevent student from over-fitting to any task. We provide an experimental result and theoretical analysis in Fig.2 and Appendix SectionA.2.

## 4 EXPERIMENTS

### 4.1 IMPLEMENTATION DETAILS

YOLOX uses CSPNet (Wang et al., 2020) as its backbone, which is trained from scratch along with detection heads for 300 epochs [1]. Following the general settings of IOD task(Li et al., 2019), we replace CSPNet with pre-trained ResNet(He et al., 2016) and PVT2(Wang et al., 2021b), which are frozen during incremental learning. We keep the other components and hyper parameters of YOLOX unchanged. The modified YOLOX, denoted as **ilYOLOX**, is used for incremental learning.

Given a leaning scenario, we continually train ilYOLOX task by task. The leaning on each task is seen as an incremental learning step. The ilYOLOX trained on old task is used as teacher to guide the next step learning of student on new task. Optimizer is SGD with warm-up iterations of 1500, a learning rate of 0.2 decayed by 10% at the $8^{th}$ and $11^{th}$ epochs respectively, a momentum of 0.9

---

[1]seen YOLOX in https://github.com/open-mmlab/mmdetection

and a weight decay of 0.0005. All experiments are performed on 8 NVIDIA 3090 GPUs with a total batch size of 16×8. All training images are randomly scaled to [416, 640] by their short sides with content shape ratios unchanged. Normalization, cutout and random horizontal flip with a probability of 50% are used for training. All test images are adaptively scaled to 640x640 and only normalized.

## 4.2 DATASETS, BENCHMARKS AND EVALUATION METRICS

We build benchmarks using Pascal VOC(Everingham et al., 2010) and MS COCO2017(Chen et al., 2015) separately, with 20 and 80 categories in each dataset. We split training and validation sets into several subsets following the alphabetic order of categories. Each split scheme is called a incremental learning scenario. For example, the scenario of COCO(40+20+20) indicates splitting COCO into three subsets with 40, 20 and 20 categories in each. Incremental learning will be carried out sequentially on these subsets step by step. At each step, only categories in new subset is loaded for training, while all previously learned categories are loaded for testing. The total learning steps equals to the number of subsets. Learning by loading all categories at once refers to joint learning. In addition, we also use the benchmark of OWOD task, which combines VOC and COCO together, abbreviated as VOCO. The first subset consists of 20 categories from VOC, while the other subsets consist of the remaining 60 categories from COCO. These categories are sorted by semantic drift.

Our benchmarks include (i)Two-Step scenarios: 40+40, 50+30, 60+20 and 70+10 on COCO; 10+10, 15+5, 19+1 on VOC. (ii)Three-Step scenarios:40+20+20 on COCO. (iii)Four-Step scenarios:20+20+20+20 on COCO and VOCO; 5+5+5+5 on VOC. (iv)Five-Step scenario:40+10+10+10+10 on COCO; 15+1+1+1+1+1 on VOC. We denote A(a-b) as the first-step normal training for categories a-b, while +B(c-d) as incremental training for categories c-d. Therefore, the scenario of COCO(40+20+20) contains three learning steps A(1-40),+B(50-60),+B(70-80).

**Evaluation Metrics.** (1) The standard COCO protocols (**mAP**) and VOC protocols ($mAP@0.5$) are used to evaluate object detection performance. The $mAP$ of joint learning and incremental learning are denoted as $mAP_{joint}$ and $mAP_{incre}$. In order to evaluate the incremental learning better, we use the following metrics. (2) **AbsGap** and **RelGap**(Menezes et al., 2022), defined as Eq.11, respectively evaluate the absolute gap and relative gap between incremental learning and joint learning at every step without cumulation. (3) **Omega** ($\Omega$)(Hayes et al., 2018; Menezes et al., 2022), defined as Eq.14(b), is used to evaluate the cumulative capability of multi-step incremental learning step by step. Similar to COCO protocols, $\Omega$ can be extended as $\Omega_{all}, \Omega_{50}, \Omega_{75}, \Omega_S, \Omega_M$ and $\Omega_L$. (4) **SDR** and **PDR** (Menezes et al., 2022), defined in Eq.12 and Eq.13, refer to the stability deficits rate on old categories and the plasticity deficits rate on new categories, respectively. (5) **SPDR**, defined in Eq.14(a), refers to the total deficits rate of stability and plasticity.

$$\textbf{(a)}AbsGap = mAP_{joint,t} - mAP_{incre,t} \quad \textbf{(b)}RelGap = \frac{mAP_{joint,t} - mAP_{incre,t}}{mAP_{joint,t}} \tag{11}$$

$$SDR = \frac{1}{N_{old}} \sum_{i=1}^{N_{old}} \frac{mAP_{joint,i} - mAP_{incre,i}}{mAP_{joint,i}} \tag{12}$$

$$PDR = \frac{1}{N_{new}} \sum_{i=N_{old}+1}^{N_{new}} \frac{mAP_{joint,i} - mAP_{incre,i}}{mAP_{joint,i}} \tag{13}$$

$$\textbf{(a)}SPDR = SDR + PDR \qquad \textbf{(b)}\Omega = \frac{1}{T} \sum_{t=1}^{T} \frac{mAP_{incre,t}}{mAP_{joint,t}} \tag{14}$$

where $T$ and $t$ refers to the total learning steps and the $t^{th}$ learning step. $i$ refers to the $i^{th}$ category. $mAP_{incre,t}$ and $mAP_{joint,t}$ refers to incremental and joint learning on the testing data of all learned categories after the $t^{th}$ learning step. $N_{old}$ and $N_{new}$ are the total number of old and new categories.

## 4.3 OVERALL PERFORMANCE

### 4.3.1 INCREMENTAL LEARNING ABILITY

Table1 reports the incremental learning results of Two-Step scenarios on COCO. Table2 reports the incremental learning results of Four-Step scenario on VOC. Compared with previous works,

including LwF (Li & Hoiem, 2018), ILOD (Shmelkov et al., 2017), CIFRCN (Hao et al., 2019), RILOD (Li et al., 2019), SID (Peng et al., 2021) and the latest best method ERD (Feng et al., 2022), our method achieves best performance under all these scenarios and all evaluation metrics. For the Four-Step scenario of VOC(5+5+5+5), our method also shows obvious performance improvement. For the most difficult scenario of Five-Step COCO(40+10+10+10+10) in Table3, our method shows overwhelming advantages over ERD under final mAP (34.06% vs 20.70%), AbsGAP (6.15% vs 19.50%), RelGAP (15.28% vs 48.51%) and $\Omega_{all}$ (0.933 vs 0.796). We further plot $\Omega_{all}$ curves in Fig.3 to highlight our advantages. $AbsGap$ and $RelGap$ reflect the knowledge distillation ability at each current step. $\Omega_{all}$ reflects the accumulated learning ability step by step, therefore reveals accumulated knowledge forgetting. Obviously, our method exhibits superior long-range incremental learning abilities. More results on COCO and VOC are shown in Appendix Tables 14 16 12. These results fully demonstrate excellent incremental learning capacity of our methods.

Table 1: Incremental learning results under different Two-Step scenarios of COCO. $mAP$ refers to the final $mAP$ of incremental learning. Upper refers to the $mAP$ of normal learning on all classes.

| Scenarios | Method | AbsGap↓ | RelGap↓ | $\Omega_{all}$ ↑ | $\Omega_{50}$ ↑ | $\Omega_{75}$ ↑ | $\Omega_S$ ↑ | $\Omega_M$ ↑ | $\Omega_L$ ↑ | mAP↑ | Upper |
|---|---|---|---|---|---|---|---|---|---|---|---|
| 40 classes + 40 classes | LwF | 23.00 | 57.21% | 0.714 | 0.718 | 0.713 | 0.670 | 0.709 | 0.733 | 17.20 | 40.20 |
| | RILOD | 10.30 | 25.62% | 0.872 | 0.886 | 0.867 | 0.841 | 0.874 | 0.888 | 29.90 | 40.20 |
| | SID | 6.20 | 15.42% | 0.923 | 0.941 | 0.916 | 0.897 | 0.935 | 0.930 | 34.00 | 40.20 |
| | ERD | 3.30 | 8.21% | 0.959 | 0.967 | 0.954 | 0.959 | 0.958 | 0.955 | 36.90 | 40.20 |
| | Ours | **2.65** | **6.58%** | **0.967** | **0.966** | **0.961** | **0.954** | **0.980** | **0.965** | **37.57** | 40.21 |
| 50 classes + 30 classes | LwF | 35.20 | 87.56% | 0.562 | 0.581 | 0.553 | 0.608 | 0.576 | 0.555 | 5.00 | 40.20 |
| | RILOD | 11.70 | 29.10% | 0.854 | 0.870 | 0.846 | 0.832 | 0.858 | 0.864 | 28.50 | 40.20 |
| | SID | 6.40 | 15.92% | 0.920 | 0.937 | 0.914 | 0.879 | 0.932 | 0.932 | 33.80 | 40.20 |
| | ERD | 3.60 | 8.96% | 0.955 | 0.963 | 0.946 | 0.918 | 0.958 | 0.960 | 36.60 | 40.20 |
| | Ours | **1.50** | **3.74%** | **0.981** | **0.971** | **0.980** | **0.974** | **0.992** | **0.972** | **38.71** | 40.21 |
| 60 classes + 20 classes | LwF | 34.40 | 85.57% | 0.572 | 0.593 | 0.561 | 0.586 | 0.596 | 0.574 | 5.80 | 40.20 |
| | RILOD | 14.80 | 36.82% | 0.816 | 0.833 | 0.807 | 0.800 | 0.829 | 0.823 | 25.40 | 40.20 |
| | SID | 7.50 | 18.66% | 0.907 | 0.927 | 0.897 | 0.871 | 0.926 | 0.917 | 32.70 | 40.20 |
| | ERD | 4.40 | 10.95% | 0.945 | 0.954 | 0.940 | 0.944 | 0.947 | 0.945 | 35.80 | 40.20 |
| | Ours | **1.87** | **4.64%** | **0.977** | **0.969** | **0.975** | **0.960** | **0.990** | **0.973** | **38.34** | 40.21 |
| 70 classes + 10 classes | LwF | 33.10 | 82.34% | 0.588 | 0.606 | 0.580 | 0.603 | 0.608 | 0.596 | 7.10 | 40.20 |
| | RILOD | 15.70 | 39.05% | 0.805 | 0.825 | 0.795 | 0.806 | 0.811 | 0.821 | 24.50 | 40.20 |
| | SID | 7.40 | 18.41% | 0.908 | 0.920 | 0.901 | 0.869 | 0.918 | 0.926 | 32.80 | 40.20 |
| | ERD | 5.30 | 13.18% | 0.934 | 0.945 | 0.929 | 0.903 | 0.940 | **0.936** | 34.90 | 40.20 |
| | Ours | **2.98** | **7.41%** | **0.963** | **0.961** | **0.958** | **0.945** | **0.975** | 0.944 | **37.23** | 40.21 |

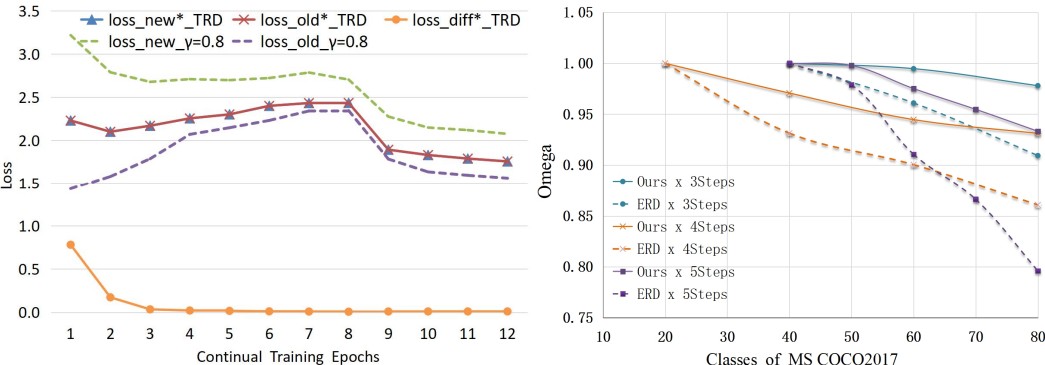

Figure 2: The training loss of old and new tasks at $\gamma = 0.8$ and TRD for Table8 respectively.

Figure 3: The $\Omega_{all}$ performance on multi-step incremental learning on MS COCO.

Table 2: Incremental learning results on the Four-Step scenario of VOC(5+5+5+5).

| Method | mAP | | | | Final mAP↑ | AbsGap↓ | RelGap↓ | $\Omega_{50}$ ↑ | Upper |
|---|---|---|---|---|---|---|---|---|---|
| | A (1-5) | +B(6-10) | +B(11-15) | +B(16-20) | | | | | |
| CF | 1.25 | 2.34 | 3.12 | 36.32 | 11.31 | 43.94 | 61.37% | 0.636 | 70.64 |
| SID | 27.26 | 40.10 | 43.02 | 34.44 | 36.21 | 35.40 | 49.43% | 0.736 | 71.60 |
| ILOD | 29.55 | 43.47 | 46.65 | 37.34 | 39.25 | 30.55 | 43.76% | 0.755 | 69.80 |
| CIFRCN | 34.60 | 44.10 | 55.60 | 59.60 | 48.48 | 22.04 | 31.25% | 0.797 | 70.51 |
| ERD | **41.25** | 57.38 | 63.57 | 53.12 | 53.83 | 16.77 | 23.55% | 0.902 | 70.60 |
| Ours | 38.04 | **61.14** | **69.03** | **53.96** | **55.54** | **15.10** | **21.37%** | **0.920** | 70.64 |

Table 3: Incremental learning results on the Five-Step scenario of COCO(40+10+10+10+10).

| | A(1-40) | | | | | | | | | |
|---|---|---|---|---|---|---|---|---|---|---|
| Method | +B(40-50) | | | | | +B(50-60) | | | | |
| | mAP↑ | AbsGap↓ | RelGap↓ | $\Omega_{all}$ ↑ | Upper | mAP↑ | AbsGap↓ | RelGap↓ | $\Omega_{all}$ ↑ | Upper |
| CF | 5.80 | 32.20 | 84.74% | 0.576 | 38.00 | 5.70 | 34.10 | 85.68% | 0.432 | 39.80 |
| RILOD | 25.40 | 12.60 | 33.16% | 0.834 | 38.00 | 11.20 | 28.60 | 71.86% | 0.650 | 39.80 |
| SID | 34.60 | 3.40 | 8.95% | 0.955 | 38.00 | 24.10 | 15.70 | 39.45% | 0.839 | 39.80 |
| ERD | 36.40 | 1.60 | 4.21% | 0.979 | 38.00 | 30.80 | 9.00 | 22.61% | 0.911 | 39.80 |
| Ours | **39.16** | **0.16** | **0.41%** | **0.998** | **39.32** | **35.97** | **2.72** | **7.03%** | **0.975** | **38.69** |
| Method | +B(60-70) | | | | | +B(70-80) | | | | |
| | mAP↑ | AbsGap↓ | RelGap↓ | $\Omega_{all}$ ↑ | Upper | mAP↑ | AbsGap↓ | RelGap↓ | $\Omega_{all}$ ↑ | Upper |
| CF | 6.30 | 29.40 | 82.35% | 0.368 | 35.70 | 3.30 | 36.90 | 91.79% | 0.311 | 40.20 |
| RILOD | 10.50 | 25.20 | 70.59% | 0.561 | 35.70 | 8.40 | 31.80 | 79.10% | 0.491 | 40.20 |
| SID | 14.60 | 21.10 | 59.10% | 0.731 | 35.70 | 12.60 | 27.60 | 68.66% | 0.648 | 40.20 |
| ERD | 26.20 | 9.50 | 26.61% | 0.866 | 35.70 | 20.70 | 19.50 | 48.51% | 0.796 | 40.20 |
| Ours | **34.22** | **4.08** | **10.65%** | **0.955** | **38.30** | **34.06** | **6.15** | **15.28%** | **0.933** | **40.21** |

### 4.3.2 COMPARED WITH OWOD METHODS

Within the OWOD paradigm, at each learning step, a model learns to detect a given set of known objects while simultaneously being capable of identifying unknown objects. These flagged unknowns can be labeled by human annotators as newly added classes for the next step learning. Given the data of these new classes, the model would continue updating its knowledge in an incremental fashion without retraining from scratch on the previously known classes. The difference between OWOD and IOD is that the former need to locate unknown objects by model itself at each step, while the latter treats all unknown objects as background until the next learning step. We compare our method with current three OWOD methods, including(Joseph et al., 2021a), SemTopology(Yang et al., 2021b) and OW-DETR(Gupta et al., 2022). The results are shown in Table4. Our method performs better than current OWOD methods under the metrics of $mAP$, $AbsGap$, $RelGap$, $\Omega_{all}$.

### 4.3.3 BALANCE LEARNING ABILITY

The stability of old knowledge and plasticity of new knowledge are usually considered as two struggle aspects for incremental learning(Menezes et al., 2022). The metrics of $SDR$, $PDR$ and $SPDR$ measure the deficits rate of stability, the deficits rate of plasticity and their combination. The lower the deficits rate, the better the stability and plasticity. $SPDR$ reflects the struggle between these two aspects. Table5 and Appendix Table14 report the balance learning results on COCO and VOC, respectively. These results reveal that our method make a better trade-off between stability and plasticity to achieve an optimal comprehensive performance. Meanwhile, although different methods shows different stability ($SDR$) and plasticity ($PDR$), the final incremental learning performance ($mAP$, $RelGap$, $\Omega$) under these scenarios shows a clear positive correlation with the $SPDR$.

## 5 ABLATION STUDY

**The Performance of HKR and TRD under Two-Step Scenario.** Table6 shows the results of ablation experiments. The two baseline methods, denoted as Onehot and Soft, use one-hot predictions and soft predictions (logits) as teacher knowledge respectively. Then we add Hybrid Knowledge

Table 4: Incremental learning results on the Four-Step scenario of VOCO(20+20+20+20).

| Method | mAP | | | | mAP↑ | AbsGap↓ | RelGap↓ | $\Omega_{all}$ ↑ | Upper |
|---|---|---|---|---|---|---|---|---|---|
| | A(1-20) | +B(20-40) | +B(40-60) | +B(60-80) | | | | | |
| ORE | 56.34 | | | | 56.34 | 0.00 | 0.00% | 1 | 56.34 |
| | 52.37 | 25.58 | | | 38.98 | 11.13 | 22.21% | 0.889 | 50.11 |
| | | 37.77 | 12.41 | | 29.32 | 15.91 | 35.18% | 0.809 | 45.23 |
| | | | 30.01 | 13.44 | 26.66 | 16.23 | 37.84% | 0.762 | 42.89 |
| SemTopology | 56.34 | | | | 56.34 | 0.00 | 0.00% | 1 | 56.34 |
| | 53.39 | 26.49 | | | 39.94 | 10.17 | 20.30% | 0.899 | 50.11 |
| | | 38.04 | 12.81 | | 29.63 | 15.60 | 34.49% | 0.817 | 45.23 |
| | | | 30.11 | 13.31 | 25.91 | 16.98 | 39.59% | 0.764 | 42.89 |
| OW-DETR | 56.34 | | | | 56.34 | 0.00 | 0.00% | 1 | 56.34 |
| | **53.55** | 33.45 | | | **42.92** | 7.19 | 14.35% | 0.928 | 50.11 |
| | | **38.25** | 15.82 | | 30.77 | 14.46 | 31.97% | 0.846 | 45.23 |
| | | | 31.38 | 17.14 | 27.82 | 15.07 | 35.14% | 0.796 | 42.89 |
| Ours | 53.90 | | | | 53.90 | 0.00 | 0.00% | 1 | 53.90 |
| | 44.51 | **37.87** | | | 41.19 | **6.62** | **13.85%** | **0.931** | 47.81 |
| | | 34.72 | **31.86** | | **33.77** | **9.72** | **22.36%** | **0.879** | 43.49 |
| | | | **32.35** | **24.00** | **30.26** | **10.91** | **26.49%** | **0.843** | 41.17 |

Table 5: Stability deficits rate(SDR), plasticity deficits rate(PDR) and their total deficits rate (SPDR), which reflect the balance learning ability between old knowledge and new knowledge.

| Scenarios | Method | SPDR↓ | SDR↓ | PDR↓ | Incremental mAP | | | Joint mAP | | |
|---|---|---|---|---|---|---|---|---|---|---|
| | | | | | Total↑ | Old | New | Total | Old | New |
| 40 classes + 40 classes | ERD | 16.66 | 8.37 | 8.29 | 36.90 | 41.60 | 32.10 | 40.20 | 45.40 | 35.00 |
| | Ours | **12.76** | **9.28** | **3.48** | **37.57** | 36.27 | 38.86 | 40.21 | 39.98 | 40.26 |
| 50 classes + 30 classes | ERD | 17.04 | 9.74 | 7.30 | 36.60 | 38.00 | 34.30 | 40.20 | 42.10 | 37.00 |
| | Ours | **3.49** | **7.97** | **-4.48** | **38.71** | 37.07 | 41.44 | 40.21 | 40.28 | 39.66 |
| 60 classes + 20 classes | ERD | 16.57 | 13.69 | 2.88 | 35.80 | 35.30 | 37.10 | 40.20 | 40.90 | 38.20 |
| | Ours | **4.46** | **7.51** | **-3.05** | **38.3**4 | 37.17 | 41.86 | 40.21 | 40.19 | 40.62 |
| 70 classes + 10 classes | ERD | 19.45 | 14.18 | 5.26 | 34.90 | 35.70 | 28.80 | 40.20 | 41.60 | 30.40 |
| | Ours | **7.22** | **9.09** | **-1.87** | **37.23** | 36.25 | 44.07 | 40.21 | 39.88 | 43.26 |

Representation (HKR) module and Task Regularized Distillation (TRD) module to the Soft baseline respectively, whose results are denoted as Soft+HKR and Soft+TRD. Finally, we add both HKR and TRD to the Soft baseline simultaneously, whose results are denoted as Soft+HKR+TRD. The results under two scenarios all show that soft knowledge is better than hard knowledge, but both are inferior to hybrid knowledge. Compared with the baseline, Soft+TRD shows higher performance improvement than Soft+HKR. This demonstrates that, as two independent components, both HKR and TRD have their own effects. Meanwhile, the results of 'Soft+HKR+TRD' get further significant improvement, demonstrating that HKR and TRD have good additivity and compatibility.

**The Performance of HKR and TRD under Multi-Step Scenario.** We make additional ablation studies under Three-Step scenario of COCO(40+20+20). The results shown in Table7 demonstrate the effectiveness of HKR and TRD clearly. It reflects that HKR can effectively take the advantages of both soft knowledge and hard knowledge in adaptive manner. TRD can further improve performance by re-balancing the old and new tasks during multi-steps incremental learning.

**The Analysis of Task Balance During Incremental Learning.** We make experiments by changing the task balancing factor ($\gamma$ in Eq.6). The results are shown in Table8, Where the optimal and suboptimal values are represented by bold and underlined digits, respectively. On one hand, when $\gamma$ changes from 0.2 to 3.0, the mAP of 'Old 70 Classes', 'New 10 Classes' and 'Final 80 classes' shows noticeable changes. TRD method gets best performance under all other metrics at a small cost of $mAP$ on 'New 10 Classes'. This fully demonstrate that task balance factor ($\gamma$) has a significant influence on incremental learning by controlling knowledge transfer from teacher to student. Especially, compared with different $\gamma$ values, TRD method gets the highest mAP for 'Old 70 Classes', indicating that TRD relieves knowledge forgetting to the greatest extent. On the other hand, TRD

Table 6: Incremental learning results under Two-Step scenarios for ablation study. We equip YOLOX with two knowledge selection methods (Onehot and Soft) as our baselines.

| Scenarios | 60 classes + 20 classes | | | | | 70 classes + 10 classes | | | | |
|---|---|---|---|---|---|---|---|---|---|---|
| Methods | $AbsGap \downarrow$ | $RelGap \downarrow$ | $\Omega_{all} \uparrow$ | $\Omega_{50} \uparrow$ | $\Omega_{75} \uparrow$ | $AbsGap \downarrow$ | $RelGap \downarrow$ | $\Omega_{all} \uparrow$ | $\Omega_{50} \uparrow$ | $\Omega_{75} \uparrow$ |
| Onehot | 3.86 | 11.27% | 0.944 | 0.948 | 0.939 | 4.91 | 14.33% | 0.928 | 0.937 | 0.926 |
| Soft | 3.29 | 9.60% | 0.952 | 0.951 | 0.954 | 4.49 | 13.10% | 0.935 | 0.939 | 0.933 |
| Soft+HKR | 3.10 | 9.04% | 0.955 | 0.951 | 0.958 | 4.19 | 12.22% | 0.939 | 0.942 | 0.941 |
| Soft+TRD | 2.95 | 8.62% | 0.957 | 0.954 | 0.958 | 4.06 | 11.83% | 0.941 | 0.943 | 0.944 |
| Soft+HKR+TRD | **2.49** | **7.26%** | **0.964** | **0.958** | **0.969** | **3.14** | **9.16%** | **0.954** | **0.953** | **0.956** |

Table 7: Ablation results under Three-Step scenario of COCO(40+20+20).

| Scenarios | A(1-40) | | | | | | | | | |
|---|---|---|---|---|---|---|---|---|---|---|
| | +B(50-60) | | | | | +B(70-80) | | | | |
| Methods | $AbsGap\downarrow$ | $RelGap\downarrow$ | $\Omega_{all} \uparrow$ | $\Omega_{50} \uparrow$ | $\Omega_{75} \uparrow$ | $AbsGap\downarrow$ | $RelGap\downarrow$ | $\Omega_{all} \uparrow$ | $\Omega_{50} \uparrow$ | $\Omega_{75} \uparrow$ |
| Onehot | 2.04 | 6.05% | 0.970 | 0.972 | 0.969 | 3.83 | 11.16% | 0.943 | 0.945 | 0.942 |
| Soft | 1.81 | 5.38% | 0.973 | 0.971 | 0.975 | 3.39 | 9.91% | 0.949 | 0.945 | 0.949 |
| Soft+HKR | 1.62 | 4.81% | 0.976 | 0.970 | 0.982 | 3.08 | 8.97% | 0.954 | 0.948 | 0.958 |
| Soft+HKR+TRD | **0.78** | **2.32%** | **0.988** | **0.980** | **0.997** | **1.60** | **4.68%** | **0.976** | **0.966** | **0.979** |

shows much more balancing ability between old and new tasks with the minimum $RSPD$. We plot the training losses of old and new tasks at $\gamma = 0.8$ and TRD in Fig.2. We can see that the $loss\_new$ is always larger than $loss\_old$ at $\gamma = 0.8$ during the entire incremental learning, which is more conducive to learning new tasks. While $loss\_new^*$ and $loss\_old^*$ are always equal to each other, and the $loss\_diff^*$ quickly approaching zero. Obviously, TRD effectively balance knowledge transfer to get the best comprehensive result by introducing task-based regularization.

Table 8: Incremental learning results on Two-Step scenario for ablation study. Hyper parameter $\gamma$ is the task balance factor (seen in Eq.6). TRD is task regularized distillation method (seen in Eq.10)

| Methods | $mAP \uparrow$ | | | $\Omega_{all} \uparrow$ | $\Omega_{50} \uparrow$ | $\Omega_{75} \uparrow$ | $RSPD \downarrow$ |
|---|---|---|---|---|---|---|---|
| | Old 70 Classes | New 10 Classes | Final 80 Classes | | | | |
| $\gamma = 0.2$ | 27.00 | 37.41 | 28.30 | 0.913 | 0.926 | 0.910 | 20.18 |
| $\gamma = 0.5$ | 28.41 | 37.70 | 29.57 | 0.932 | 0.937 | 0.931 | 15.23 |
| $\gamma = 0.6$ | 28.37 | **38.02** | 29.58 | 0.932 | 0.936 | 0.929 | 14.47 |
| $\gamma = 0.7$ | 28.58 | 37.69 | 29.72 | 0.934 | 0.939 | 0.932 | 14.76 |
| $\gamma = 0.8$ | 28.92 | 37.48 | 29.99 | 0.938 | 0.939 | 0.936 | 14.30 |
| $\gamma = 0.9$ | 28.59 | 37.53 | 29.71 | 0.934 | 0.936 | 0.933 | 15.14 |
| $\gamma = 1.0$ | 28.72 | 37.14 | 29.78 | 0.935 | 0.939 | 0.933 | 15.79 |
| $\gamma = 1.5$ | 28.61 | 36.96 | 29.66 | 0.933 | 0.936 | 0.929 | 16.60 |
| $\gamma = 2.0$ | 28.37 | 36.04 | 29.33 | 0.928 | 0.931 | 0.929 | 19.78 |
| $\gamma = 3.0$ | 27.11 | 35.14 | 28.12 | 0.910 | 0.917 | 0.907 | 25.90 |
| TRD | **29.24** | 36.97 | **30.21** | **0.941** | **0.943** | **0.944** | **13.71** |

## 6 CONCLUSION

In order to improve the performance of incremental object detection, we propose a knowledge distillation method that combines knowledge selection and transfer strategy effectively. For the first strategy, hard knowledge and soft knowledge are adaptively combined to construct a kind of hybrid knowledge representation to use teacher knowledge effectively. For the second strategy, loss difference are combined to construct task regularized distillation loss to enhance task balance learning. Extensive experiments under different scenarios validate the effectiveness of our method. Most existing methods mix feature, response and relation distillation in a complex framework to relieve catastrophic forgetting. We demonstrate that catastrophic forgetting can be significantly alleviated by reducing knowledge fuzziness of teacher and suppressing imbalance learning between old and new tasks.More experiments and analyses are provided in Appendix A.

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

# A APPENDIX

## A.1 MORE DETAILS OF IMPLEMENTATION AND EXPERIMENTS

Our experiments are implemented based on YOLOX detector (Ge et al., 2021). YOLOX is a typical one-stage anchor-free detector among famous and widely used YOLO series, which can contribute to the typical verification of our method. The official YOLOX provided in MMDetection [2] adopts a training schedule of 300 epochs from scratch with CSPDarkNet as its backbone. In order to boost its performance, official YOLOX adopts a large input image size of $1333 \times 800$ and a very strong data augmentation strategy, including Mosaic, MixUP, Photo Metric Distortion, EMA, Random Affine and so on. For economical training and stable reproducibility, we use a small input image size of 640x640 and drop these data augmentation tricks to reduce the randomness of incremental learning. To enhance model performance, we use the pre-trained PVT2-b2(Wang et al., 2021b) and ResNet50(He et al., 2016) as our backbones on MS COCO and Pascal VOC, respectively. YOLOX has multiple versions with different parameter quantities, named as YOLOX-L, YOLOX-M, YOLOX-S and so on. In order to ensure similar baseline performance with other methods under joint learning, we use YOLOX-L and YOLOX-M for MS COCO and Pascal VOC, respectively.

---

[2]seen YOLOX in https://github.com/open-mmlab/mmdetection

## A.2 ALGORITHM AND THEORETICAL ANALYSIS

**Hybrid Knowledge Representation.** Teacher outputs soft logits and one-hot labels as its predictions. Soft logits contains more information about between-class confidences and is regarded as a kind of soft knowledge (Hinton et al., 2015). Knowledge distillation methods are developed prosperously under this background in image classification and object detection tasks. However, there are significant difference between the two tasks. For image classification, since an input image contains only one instance, the image knowledge is just the knowledge of its instance. However, for object detection, an input image always contains several instances, the image knowledge is a collection of the knowledge of all instances on it. In other words, image-level knowledge should be the combination of instance-level knowledge, thus the combination mode is very important. Soft logits and one-hot labels provide two kinds of instance-level knowledge, soft knowledge and hard knowledge respectively. Soft knowledge contains confidence relations among categories, but brings knowledge fuzziness inevitably. While, hard knowledge has completely opposite effects. By combining their advantages, we construct an image-level hybrid knowledge representation, abbreviated as HKR. The ablation studies in Table6 and Table 7 demonstrate its good performance. Algorithm.1 provides more details about HKR method.

---

**Algorithm 1** Algorithm of Hybrid Knowledge Representation

**Input**: $BatchImgs$: a batch of images from new task, batch size $N$, teacher detector $\theta^T$
**Output**: $Hybrid$: the hybrid predictions of teacher outputs

1: Inference $BatchImgs$ with $\theta^T$ yields two kinds of predictions for the old categories, soft logits $SoftPred^T_{old}$ and one-hot labels $OnehotPred^T_{old}$.
2:
3: // Calculating the confidence difference (abbreviate as ConfDiff) for each image.
4: Create $ConfDiff$
5: **for** i, img in $BatchImgs$ **do**
6:   Compute $Conf^i_{first\_max} = max(SoftPred^T_{old,i})$
7:   Compute $Conf^i_{second\_max} = sorted(SoftPred^T_{old,i}, descending = True)[1]$
8:   Compute $ConfDiff^i = Conf^i_{first\_max} - Conf^i_{second\_max}$
9:   $ConfDiff.append(ConfDiff^i)$
10: **end for**
11:
12: // Calculating binary vector $quality$ and final hybrid prediction $Hybrid$.
13: Compute $quality = \textbf{ConfDiff} > \frac{1}{N}\sum_i^N \textbf{ConfDiff}_i$
14: Compute $Hybrid = quality \cdot OnehotPred^T_{old} + (1 - quality) \cdot SoftPred^T_{old}$

---

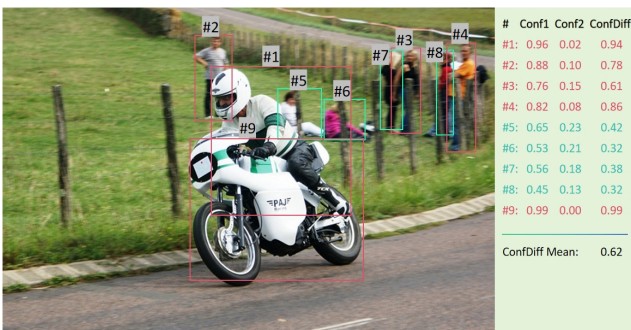

Figure 4: The illustration of HKR method. The image is inferenced by a teacher detector trained on the first 60 classes of MS COCO. Conf1 and Conf2 respectively refer to the first and second maximum confidences. ConfDiff is their conference difference. Red and green boxes will be regarded as hard knwoledge and soft knowledge, respectively.

**Task Regularized Distillation.** Regularization method is very import for statistical machine learning, which can prevent model from over-fitting to some part of data. Classical regularization methods introduce a weight constraint in terms of p-Norm as model penalties. Other regularization methods include early termination of training and soft weight sharing (Nowlan & Hinton, 1992). Dropout works by dropping some connections randomly to prevent over-fitting of deep neural networks (Srivastava et al., 2014). In incremental learning, model needs to be trained task by task in the continuous data flow, therefore bring task-based imbalance learning and leading to catastrophic knowledge forgetting of old task. A few previous works propose regularization-based methods on incremental image classification (Kirkpatrick et al., 2017; Li & Hoiem, 2018). We give a solution by proposing task-based regularized distillation loss Eq.10 for IOD tasks. It explicitly uses the loss difference between old and new tasks as a model penalty to constraint optimization process. Just like Fig.2 shows, throughout the entire incremental learning, the losses of new task and old task are equivalent ($loss\_new^*\_TRD == loss\_old^*\_TRD$, their curves coincide with each other). Table6, Table8, Table10 and Fig.5 all show its strong effectiveness to prevent incremental model from over-fitting to old and new task. Table5 and Fig.6 show that our method makes a better trade-off between the stability of old knowledge and the plasticity of new knowledge on both MS COCO and Pascal VOC. Its algorithm can be seen in Algorithm.2.

---

**Algorithm 2** Algorithm of Task Regularized Distillation

---

**Input**: Image $I$, teacher detector $\theta^T$, student detector $\theta^S$, $GtLabels_{new}$ and $GtBBoxes_{new}$ of current new task. $\alpha$=1, $\beta$=5 for YOLOX. The default value of $\eta$ is 1.

**Output**: the detection loss of $Loss^*_{total}$ for student detector.

1:
2: // Calculating the detection loss $Loss_{new}$ of student detector on new task.
3: // Calculating $Loss^{new}_{cls}$ and $Loss^{new}_{loc}$ by classification loss function and location loss function of the original base detector, respectively.
4: Inference $I$ with $\theta^S$ yields the logits predictions $SoftPred_{new}$ and bounding-box predictions $BBoxPred_{new}$ for new categories of the current task.
5: Compute $Loss^{new}_{cls} = CrossEntropyLoss(GtLabels_{new}, SoftPred_{new})$
6: Compute $Loss^{new}_{loc} = IoULoss(GtBBoxes_{new}, BBoxPred_{new})$
7: Compute $Loss_{new} = \alpha \cdot Loss^{new}_{cls} + \beta \cdot Loss^{new}_{loc}$
8:
9: // Calculating the detection loss $Loss_{old}$ of student detector on old task.
10: // Calculating $Loss^{old}_{cls}$ and $Loss^{old}_{loc}$ by classification loss function and location loss function of the original base detector, respectively.
11: Inference $I$ with $\theta^S$ yields the logits predictions $SoftPred^S_{old}$ and bounding-box predictions $BBoxPred^S_{old}$ for old categories of previously learned task.
12: Inference $I$ with $\theta^T$ yields the logits predictions $SoftPred^T_{old}$ and bounding-box predictions $BBoxPred^T_{old}$ for old categories of previously learned task.
13: Compute $Loss^{old}_{cls} = CrossEntropyLoss(SoftPred^T_{old}, SoftPred^S_{old})$
14: Compute $Loss^{old}_{loc} = IoULoss(BBoxPred^T_{old}, BBoxPred^S_{old})$
15: Compute $Loss_{old} = \alpha \cdot Loss^{old}_{cls} + \beta \cdot Loss^{old}_{loc}$
16:
17: // Calculating the detection loss $Loss^*_{total}$ of student detector on old and new task.
18: // [] refers the detach operation of PyTorch, which can separate a variable from the current computed graph to remove the gradient back-propagation of that part.
19: compute $Loss^*_{old} = [\frac{2 \cdot Loss_{new}}{Loss_{old} + Loss_{new}}] \cdot Loss_{old}$
20: compute $Loss^*_{new} = [\frac{2 \cdot Loss_{old}}{Loss_{old} + Loss_{new}}] \cdot Loss_{new}$
21: compute $Loss^*_{diff} = (Loss_{old} - Loss_{new})^2$
22: compute $Loss^*_{total} = Loss^*_{new} + Loss^*_{old} + \eta \cdot Loss^*_{diff}$

---

Here we give an theoretical analysis for our TRD method. We use the definition of $Loss^*_{old}$ in Eq.7, $Loss^*_{new}$ in Eq.7, $Loss_{total}$ in Eq.6, $Loss^*_{total}$ and $Loss^*_{diff}$ in Eq.9. The $\gamma$ in $Loss_{total}$ is set as default value of 1. Considering the difference between $Loss_{total}$ and $Loss^*_{total}$, we can have the following deductions.

Firstly, considering a single forward propagation

$$
\begin{aligned}
(Loss_{old} + Loss_{new}) - (Loss^*_{old} + Loss^*_{new}) &= Loss_{old} + Loss_{new} \\
&\quad - [\frac{2 \cdot Loss_{new}}{Loss_{old} + Loss_{new}}] \cdot Loss_{old} \\
&\quad - [\frac{2 \cdot Loss_{old}}{Loss_{old} + Loss_{new}}] \cdot Loss_{new} \\
&= (Loss_{old} + Loss_{new}) - \frac{4 \cdot Loss_{old} \cdot Loss_{new}}{Loss_{old} + Loss_{new}} \\
&= \frac{Loss^2_{old} + Loss^2_{new} - 2 \cdot Loss_{old} \cdot Loss_{new}}{Loss_{old} + Loss_{new}} \\
&= \frac{(Loss_{old} - Loss_{new})^2}{Loss_{old} + Loss_{new}}
\end{aligned}
$$

Because of $Loss_{old} > 0$ and $Loss_{new} > 0$ at most of the time, there exists

$$Loss_{old} + Loss_{new} \geq Loss^*_{old} + Loss^*_{new} \tag{15}$$

if and only if $Loss_{old} == Loss_{new}$, the equal sign established.

For student detector, the traditional $Loss_{total}$ and our $Loss^*_{total}$ are defined as following

$$
\begin{aligned}
Loss_{total} &= Loss_{new} + Loss_{old} \\
Loss^*_{total} &= Loss^*_{new} + Loss^*_{old} + \eta \cdot Loss^*_{diff}
\end{aligned}
$$

Then we have

$$Loss^*_{total} - Loss_{total} = \eta \cdot (Loss_{old} - Loss_{new})^2 - \frac{(Loss_{old} - Loss_{new})^2}{Loss_{old} + Loss_{new}} \tag{16}$$

When gradient back-propagation gradually reduce the $Loss^*_{total}$, the additional item $Loss^*_{diff}$ with a default coefficient of $\eta = 1$ will reach down to zero. In other words, $Loss_{old}$ will be gradually equal to $Loss_{new}$. Then the $Loss^*_{total}$ will be gradually reach up to $Loss_{total}$. Obviously, if there is no item of $Loss^*_{diff}$, then $Loss^*_{total}$ will always be smaller than $Loss_{total}$, which easily leads to insufficient training and under-fitting. Therefore, with the help of $Loss^*_{diff}$, TRD method can realize sufficient training on both old and new tasks. With the help of $Loss^*_{new}=Loss^*_{old}$, TRD method can affect optimization process and thereby contributing to prevent detector from over-fitting to any task.

The experiments in Fig.2 effectively support the above theoretical analysis. During the entire incremental training, $Loss_{new}$ is always greater than $Loss_{old}$. For the object function $Loss_{total}$, it means that the new data brings in a significant and persistent impact on student detector, implying that detector trends to over-fitting on new task and occurs knowledge forgetting of old task. While for TRD object function $Loss^*_{total}$, $Loss^*_{diff}$ gradually and quickly reduced to a very small value (seen the yellow solid line in Fig.2). The imbalance impact from new data will be under control, thus effectively alleviating the over-fitting of new task and the catastrophic forgetting of old task during incremental training.

### A.3 COMPARED WITH KL DIVERGENCE LOSS

Kullback-Leibler Divergence loss (denoted as KLD loss) is used for knowledge distillation of image classification (Hinton et al., 2015). YOLOX (Ge et al., 2021) uses cross entropy loss (denoted as CE loss) for its classification head. Table9 shows the comparison results of this two losses on

incremental object detection. The experiments adopt the same loss weight setting with $\alpha = 1$ and $\beta = 5$ (seen in Eq.4 and Eq.5) for the two losses. $T$ is temperature factor, a hyper parameter of KLD loss. When the temperature $T$ changes from 1 to 5, ilYOLOX gets its best performance (marked by brown color) under the medium temperature of 3. However, CE loss has a much better performance than KLD loss (marked by underline). Our TRD loss exceeds both KLD and CE loss. The use of KLD loss usually requires careful adjustment of temperature factor $T$ and loss weight $\alpha$. However, the change of $\alpha$ in $Loss_{old}$ will destroy the loss consistency about classification and location between old task ($Loss_{old}$, Eq.4) and new task ($Loss_{new}$, Eq.5), which will influence task balance during incremental learning. Based on this consideration and the experiment results in Table9, we use cross entropy loss as our fundamental knowledge transfer strategy.

Table 9: The comparison with KL Divergence Loss

| Scenarios | 70 classes + 10 classes | | | | | |
|---|---|---|---|---|---|---|
| Methods | $mAP\uparrow$ | | | $AbsGap\downarrow$ | $RelGap\downarrow$ | $\Omega_{all}\uparrow$ |
| | Old 70 Classes | New 10 Classes | Final | | | |
| KLD T=1 | 25.55 | 35.19 | 26.75 | 7.51 | 21.92% | 0.890 |
| KLD T=2 | 26.36 | 35.87 | 27.55 | 6.71 | 19.59% | 0.902 |
| KLD T=3 | 26.86 | 36.29 | 28.04 | 6.22 | 18.16% | 0.909 |
| KLD T=4 | 26.84 | 35.96 | 27.98 | 6.29 | 18.35% | 0.908 |
| KLD T=5 | 26.49 | 35.50 | 27.62 | 6.65 | 19.41% | 0.903 |
| CE Loss | 28.72 | **37.14** | 29.78 | 4.49 | 13.10% | 0.935 |
| TRD Loss | **29.24** | 36.97 | **30.02** | **4.24** | **12.38%** | **0.938** |

## A.4 CATASTROPHIC FORGETTING CAUSED BY IMBALANCED LEARNING

In order to further analyses the influence of task balance, we conduct experiments on a small dataset. The dataset consists of 3800 images, 9 classes (commonly seen toys including car, truck, train, person and so on) and have $400 \sim 500$ instances for every class with a relative balanced category distribution. We build a Two-Step scenario of 5 classes + 4 classes as our incremental object detection benchmark. The experiment results are illustrated in Fig.5 and Table10. Fig.5 shows that even a very small change of the task balance factor $\gamma$ (defined in Eq.6) from 1 to 0.9 can lead to dramatically descending of the mAP of old task (blue dotted curve). It fully demonstrates that the loss imbalance between old and new tasks can bring significant catastrophic forgetting during incremental leaning. In Table10, when $\gamma$ is adjusted from 1.3 to 0.7, the mAP of old task ('Old 5 Classes') drops from 53.58% to 5.16%. Among all manually set gamma values, $\gamma = 1.0$ achieved good balance and performance. However, it is significant inferior to our TRD and TRD+HKR methods. This supplementary experiment further demonstrates the importance of balancing old and new tasks for IOD and the effectiveness of our methods.

Table 10: Task balance experiment on a small dataset

| Scenarios | 5 classes + 4 classes | | | | | |
|---|---|---|---|---|---|---|
| Methods | mAP↑ | | | AbsGap↓ | RelGap↓ | $\Omega_{all}\uparrow$ |
| | Old 5 Classes | New 4 Classes | Final | | | |
| $\gamma = 0.7$ | 5.16 | 58.73 | 28.97 | 32.31 | 52.73% | 0.736 |
| $\gamma = 0.9$ | 5.10 | 59.20 | 29.14 | 32.14 | 52.44% | 0.738 |
| $\gamma = 1.0$ | 54.24 | 60.18 | 56.88 | 4.40 | 7.18% | 0.964 |
| $\gamma = 1.1$ | 54.34 | 59.98 | 56.84 | 4.44 | 7.25% | 0.964 |
| $\gamma = 1.3$ | 53.58 | 59.05 | 56.01 | 5.27 | 8.60% | 0.957 |
| TRD | 54.46 | 64.18 | 58.78 | 2.50 | 4.08% | 0.980 |
| TRD+HKR | **54.66** | **64.63** | **59.09** | **2.19** | **3.58%** | **0.982** |

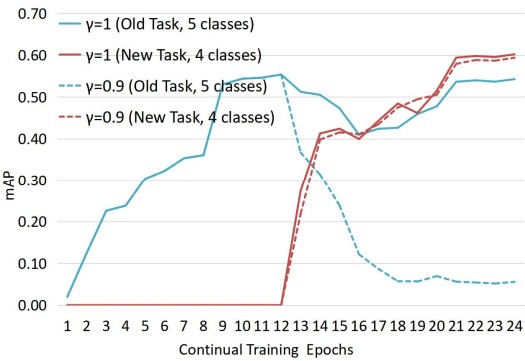

Figure 5: The catastrophic forgetting caused by imbalance learning of old and new tasks.

## A.5   GENERALITY STUDIES ON OTHER DETECTORS

Most methods that rely on feature distillation require adaptation to the feature layers of the original detector. This limits their generality to some extent. Benefit from its isolated design, our method has good generality. Our method includes two modules, HKR and TRD, which does not rely on the network architecture of original detectors. HKR mixes soft and hard knowledge by analyzing the confidence of teacher's outputs, thereby enhancing knowledge selection. TRD balances the losses between new and old tasks, preventing the model from overfitting one of them, thereby enhancing knowledge transfer. During entire incremental learning, HKR and TRD only rely on the outputs and losses, therefore they does not require much invasive adaptation to the original detector.

To validate the generality of our method, we perform experiments on a Transformer detector AdaMixer(Gao et al., 2022). For AdaMixer, we only need to replace Cross-Entropy loss with Focal lossLin et al. (2017) as classification loss ($Loss_{cls}$ in Eq.4 and Eq.5), and replace the IOU loss with GIoU Loss as location loss ($Loss_{loc}$ in Eq.4 and Eq.5). Meanwhile, let the balance factors of $\alpha$ and $\beta$ be the same with the official AdaMixer implementation[3]. Other settings are consistent with section 4.1. We only need to adjust our method slightly for adapting the outputs and losses of different detectors. Results in Table11 shows that our method still brings stable gain compared with previous best method ERD(Feng et al., 2022), which indicates its good generalization ability.

Table 11: Incremental learning results under different Two-Step scenarios of COCO.

| Scenarios | Method | AbsGap↓ | RelGap↓ | $\Omega_{all}$ ↑ | $\Omega_{50}$ ↑ | $\Omega_{75}$ ↑ | $\Omega_S$ ↑ | $\Omega_M$ ↑ | $\Omega_L$ ↑ |
|---|---|---|---|---|---|---|---|---|---|
| 40 classes + 40 classes | ERD | 3.30 | 8.21% | 0.959 | 0.967 | 0.954 | 0.959 | 0.958 | 0.955 |
| | Ours(AdaMixer) | **2.55** | **6.76%** | **0.963** | **0.969** | **0.958** | **0.955** | **0.968** | **0.966** |
| 50 classes + 30 classes | ERD | 3.60 | 8.96% | 0.955 | 0.963 | 0.946 | 0.918 | 0.958 | 0.960 |
| | Ours(AdaMixer) | **2.45** | **6.12%** | **0.962** | **0.968** | **0.959** | **0.932** | **0.966** | **0.970** |
| 60 classes + 20 classes | ERD | 4.40 | 10.95% | 0.945 | 0.954 | 0.940 | 0.944 | 0.947 | 0.945 |
| | Ours(AdaMixer) | **3.41** | **8.92%** | **0.952** | **0.963** | **0.944** | **0.950** | **0.956** | **0.950** |
| 70 classes + 10 classes | ERD | 5.30 | 13.18% | 0.934 | 0.945 | 0.929 | 0.903 | 0.940 | **0.936** |
| | Ours(AdaMixer) | **2.98** | **7.41%** | **0.963** | **0.961** | **0.958** | **0.945** | **0.975** | 0.944 |

## A.6   MORE EXPERIMENTS ON MS COCO

We conduct additional experiments on MS COCO(Everingham et al., 2010) to demonstrate the effectiveness of our method. Table 12 and Table 13 respectively report the incremental learning results on COCO(40+20+20) and COCO(20+20+20+20). Table12 shows that our method get a very significant gain compared with ERD. The gains of mAP, $AbsGap$, $RelGap$ and $\Omega_{all}$ are 5.59%, -5.57%,

---

[3]https://github.com/open-mmlab/mmdetection/tree/master/configs/adamixer

Table 12: Incremental learning results under Three-Step scenario of COCO(40+20+20).

| | A(1-40) | | | | | | | | | |
|---|---|---|---|---|---|---|---|---|---|---|
| | +B(40-60) | | | | | +B(60-80) | | | | |
| | mAP | AbsGap↓ | RelGap↓ | $\Omega_{all}$ ↑ | Upper | mAP | AbsGap↓ | RelGap↓ | $\Omega_{all}$ ↑ | Upper |
| CF | 10.70 | 29.10 | 73.38% | 0.634 | 39.80 | 9.40 | 30.80 | 76.62% | 0.501 | 40.20 |
| RILOD | 27.80 | 12.00 | 30.85% | 0.849 | 39.80 | 15.80 | 24.40 | 60.70% | 0.697 | 40.20 |
| SID | 34.00 | 5.80 | 15.42% | 0.927 | 39.80 | 23.80 | 16.40 | 40.80% | 0.815 | 40.20 |
| ERD | 36.70 | 3.10 | 7.79% | 0.961 | 39.80 | 32.40 | 7.80 | 19.40% | 0.909 | 40.20 |
| Ours | **38.28** | **0.41** | **1.05%** | **0.995** | 38.69 | **37.99** | **2.23** | **5.53%** | **0.978** | 40.21 |

-13.87%, and 0.0773. Table13 shows that although our method lags behind at the leaning step of +B(20-40), it quickly leads at the leaning step of +B(40-60) and significantly expands its lead at the learning step of +B(60-80). We list the improvements at the last learning step of +B(60-80) to show our strengths. It demonstrates that our method has better long-range incremental learning ability.

Table 13: Incremental learning results under the Four-Step scenario of COCO(20+20+20+20).

| Method | mAP | | | | mAP↑ | AbsGap↓ | RelGap↓ | $\Omega_{all}$ ↑ | Upper |
|---|---|---|---|---|---|---|---|---|---|
| | A (1-20) | +B(20-40) | +B(40-60) | +B(60-80) | | | | | |
| ERD | **34.58** | | | | 34.58 | 0.00 | 0.00% | 1 | 34.58 |
| | 30.55 | 41.23 | | | 35.89 | 5.71 | 13.73% | 0.931 | 41.60 |
| | 27.07 | 37.52 | 35.53 | | 33.37 | 6.43 | 16.16% | 0.900 | 39.80 |
| | 22.33 | 32.10 | 29.41 | 35.60 | 29.86 | 10.34 | 25.72% | 0.861 | 40.20 |
| Ours | 34.30 | | | | 34.30 | 0.00 | 0.00% | 1 | 34.30 |
| | **31.50** | **42.29** | | | **36.89** | **2.29** | **5.84%** | **0.971** | 39.18 |
| | **30.60** | **39.89** | **36.60** | | **34.54** | **4.16** | **10.74%** | **0.945** | 38.69 |
| | **29.68** | **39.93** | **35.58** | **38.05** | **35.81** | **4.41** | **10.95%** | **0.931** | 40.21 |

## A.7 MORE EXPERIMENTS ON PASCAL VOC

**Dataset and Benchmarks**. We conduct experiments on Pascal VOC(Everingham et al., 2010) to demonstrate the effectiveness of our method. The total training set of VOC2007 and VOC2012 contains 16551 images and 47223 instances. The test set of VOC2007 contains 4952 images and 14976 instances. By combining VOC2007 and VOC2012, we build different incremental learning scenarios based on the total training set and test set, including VOC(10+10), VOC(15+5), VOC(19+1), VOC(5+5+5+5), VOC(15+1+1+1+1+1). Other details are the same as section.4.2.

**Incremental Learning Ability**. We compare our method with most of the previous methods on the Two-Step VOC scenarios, which are shown in Tabel.14 and Table 17. These methods cover a variety of technical routes, including knowledge distillation, parameter isolation, examplar-replay, pseudo-labels and meta-learning. The results in Table14 show that our method gets the best performance under the scenarios of VOC(10+10), as well as competitive performance under the scenario of VCO(15+5) and VOC(19+1). The $AbsGap$, $RelGap$, $\Omega_{50}$ and $SPDR$ all get ideal and consistent results. Meanwhile, the final mAP (denoted as Total Incremental mAP) also gets very good values under these scenarios. Table15 shows the incremental learning results on the Four-Step scenario of VOC(5+5+5+5). Table16 shows the incremental learning results on the Six-Step scenario of VOC(15+1+1+1+1+1). Compared with other methods, our method gets better performance at each learning step, demonstrating its better learning ability on long task sequences.

**Balancing Learning Ability**. Considering the stability of old knowledge and the plasticity of new knowledge, we plot the relation curves among $SDR$, $PDR$, $SPDR$ and $RelGap$ in Fig.6 to make the intuitive observation. In numerical terms, $SPDR = SDR + PDR$, seen in Eq.14(a).

In both Fig.6(a) and Fig.6(b), the incremental learning performance ($RelGap$) reflects positive correlation with $SPDR$. However there are notable differences between Fig.6(a) and Fig.6(b). On one

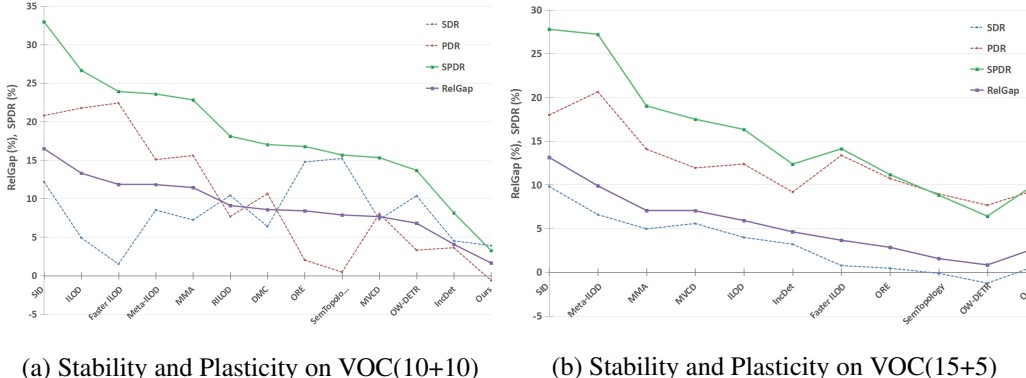

(a) Stability and Plasticity on VOC(10+10)      (b) Stability and Plasticity on VOC(15+5)

Figure 6: Incremental learning performance ($RelGap$) on VOC benchmarks, the stability deficits rate ($SDR$) of old knowledge, the plasticity deficits rate ($PDR$) of new knowledge and their total deficits rate ($SPDR$). (a) results on VOC(10+10). (b) results on VOC(15+5).

hand, some previous methods have contrary performance of stability and plasticity on VOC(10+10). They sacrifice one in exchange for the promotion of the other. For example, OWOD methods (ORE, SemTopology, OW-DETR) obtain better plasticity at the cost of stability, while ILOD-based methods (ILOD, Faster ILOD and Meta-ILOD) do the opposite. Other methods (RILOD, DMC, MVCD, IncDet and Ours) make a better trade-off between stability and plasticity, and our method do best. On the other hand, all methods show consistent phenomenon on VOC(15+5), in which stability far exceeds plasticity. In the scenario of VOC(15+5), old knowledge and new knowledge respectively contains 15 and 5 categories, therefore sacrificing the latter has less impact on final performance.

Obviously, these remarkable results once again reveal that the balance between stability of old knowledge and plasticity of new knowledge is crucial to incremental learning. The total deficits rate of stability and plasticity (SPDR) reflects a strong correlation with the final incremental learning performance. Compared with most of these methods in Table17, our method performs better on both $SDR$ and $PDR$, therefore leading to better comprehensive performance on $SPDR$ as well as the best final incremental performance.

Table 14: Incremental learning results under Two-Step scenarios of VOC(10+10), VOC(15+5) and VOC(19+1). The Incremental mAP of Total, Old and New respectively refers to the evaluation performance on total, old and new classes after the $2^{th}$ learning step. Similarly, the Joint mAP of Total, Old and New respectively refers to the evaluation performance on total, old and new classes after once normal learning. Joint mAP can be seen as the Upper Bound mAP of incremental learning. The bold numbers and underlined numbers respectively refer to the best and second best values. Sec. is the abbreviation for Scenarios. The compared methods and their reference information can be found in Table 17

| Sce. | Method | AbsGap↓ | RelGap↓ | $\Omega_{50}$↑ | SPDR↓ | SDR↓ | PDR↓ | Incremental mAP | | | Joint mAP | | |
|---|---|---|---|---|---|---|---|---|---|---|---|---|---|
| | | | | | | | | Total↑ | Old | New | Total | Old | New |
| 10 + 10 | SID | 11.80 | 16.48% | 0.918 | 32.97 | 12.18 | 20.78 | 59.80 | 62.70 | 56.80 | 71.60 | 71.40 | 71.70 |
| | ILOD | 9.38 | 13.30% | 0.934 | 26.67 | 4.91 | 21.75 | 61.14 | 67.34 | 54.93 | 70.51 | 70.82 | 70.20 |
| | Faster ILOD | 8.35 | 11.84% | 0.941 | 23.90 | **1.50** | 22.41 | 62.16 | 69.76 | 54.47 | 70.51 | 70.82 | 70.20 |
| | Meta-ILOD | 8.88 | 11.81% | 0.941 | 23.58 | 8.51 | 15.07 | 66.31 | 68.36 | 64.26 | 75.19 | 74.72 | 75.66 |
| | MMA | 8.60 | 11.44% | 0.943 | 22.82 | 7.23 | 15.59 | 66.60 | 69.3 | 63.9 | 75.20 | 74.70 | 75.70 |
| | RILOD | 6.80 | 9.10% | 0.954 | 18.09 | 10.42 | 7.67 | 67.90 | 67.48 | 68.36 | 74.70 | 75.33 | 74.04 |
| | DMC | 6.40 | 8.57% | 0.957 | 17.01 | 6.37 | 10.64 | 68.30 | 70.53 | 66.16 | 74.70 | 75.33 | 74.04 |
| | ORE | 5.93 | 8.41% | 0.958 | 16.76 | 14.76 | 2.01 | 64.58 | 60.37 | 68.79 | 70.51 | 70.82 | 70.20 |
| | SemTopology | 5.55 | 7.87% | 0.961 | 15.67 | 15.21 | 0.46 | 64.96 | 60.03 | 69.88 | 70.51 | 70.80 | 70.20 |
| | MVCD | 5.48 | 7.66% | 0.962 | 15.31 | 7.29 | 8.02 | 66.09 | 66.15 | 66.02 | 71.57 | 71.35 | 71.78 |
| | OW-DETR | 4.80 | 6.81% | 0.966 | 13.67 | 10.36 | 3.30 | 65.71 | 63.48 | 67.88 | 70.51 | 70.82 | 70.20 |
| | IncDet | 3.00 | 4.07% | 0.980 | 8.14 | 4.52 | 3.62 | **70.80** | 69.70 | 71.80 | 73.80 | 73.00 | 74.50 |
| | Ours | **1.15** | **1.63%** | **0.992** | **3.26** | 3.89 | **-0.62** | 69.49 | 67.63 | 71.35 | 70.64 | 70.36 | 70.91 |
| 15 + 5 | SID | 13.70 | 20.88% | 0.896 | 41.35 | 24.49 | 16.86 | 51.90 | 52.26 | 51.54 | 65.60 | 69.21 | 61.99 |
| | Meta-ILOD | 7.42 | 9.87% | 0.951 | 27.20 | 6.58 | 20.62 | 67.77 | 71.73 | 55.90 | 75.19 | 76.78 | 70.42 |
| | MMA | 5.30 | 7.05% | 0.965 | 19.01 | 4.95 | 14.06 | 69.90 | 73.00 | 60.50 | 75.20 | 76.80 | 70.40 |
| | MVCD | 5.02 | 7.02% | 0.965 | 17.48 | 5.56 | 11.92 | 66.54 | 69.41 | 57.92 | 71.57 | 73.50 | 65.76 |
| | ILOD | 4.16 | 5.90% | 0.971 | 16.32 | 3.97 | 12.36 | 66.35 | 69.25 | 57.60 | 70.51 | 72.11 | 65.72 |
| | IncDet | 3.40 | 4.61% | 0.977 | 12.35 | 3.20 | 9.16 | **70.40** | 72.70 | 63.50 | 73.80 | 75.10 | 69.90 |
| | Faster ILOD | 2.57 | 3.64% | 0.982 | 14.11 | 0.75 | 13.36 | 67.94 | 71.57 | 56.94 | 70.51 | 72.11 | 65.72 |
| | ORE | 2.00 | 2.84% | 0.986 | 11.16 | 0.44 | 10.71 | 68.51 | 71.79 | 58.68 | 70.51 | 72.11 | 65.72 |
| | OW-DETR | 1.09 | 1.55% | 0.992 | 8.81 | -0.14 | 8.95 | 69.42 | 72.21 | 59.84 | 70.51 | 72.11 | 65.72 |
| | SemTopology | 0.58 | 0.82% | 0.996 | 6.39 | -1.26 | 7.65 | 69.93 | 73.01 | 60.69 | 70.51 | 72.10 | 65.72 |
| | Ours | **1.95** | **2.76%** | **0.986** | **10.14** | 0.72 | 9.41 | 68.69 | 71.65 | 59.82 | 70.64 | 72.17 | 66.04 |
| 19 + 1 | SID | 20.10 | 30.64% | 0.847 | 60.82 | 33.82 | 27.00 | 45.50 | 46.33 | 44.67 | 65.60 | 70.01 | 61.19 |
| | RILOD | 9.70 | 12.99% | 0.935 | 59.60 | 10.93 | 48.67 | 65.00 | 66.33 | 40.40 | 74.70 | 74.47 | 78.70 |
| | Meta-ILOD | 4.97 | 6.60% | 0.967 | 27.56 | 5.82 | 21.74 | 70.23 | 70.89 | 57.60 | 75.19 | 75.27 | 73.60 |
| | MMA | 4.50 | 5.98% | 0.970 | 19.44 | 5.58 | 13.86 | 70.70 | 71.10 | 63.40 | 75.20 | 75.30 | 73.60 |
| | DMC | 3.90 | 5.21% | 0.974 | 17.12 | 4.80 | 12.33 | **70.81** | 70.90 | 69.00 | 74.70 | 74.47 | 78.70 |
| | ILOD | 2.79 | 3.96% | 0.980 | 11.37 | 3.97 | 7.40 | 67.72 | 67.72 | 65.10 | 70.51 | 70.52 | 70.30 |
| | Faster ILOD | 1.95 | 2.77% | 0.986 | 15.37 | 2.28 | 13.09 | 68.56 | 68.91 | 61.10 | 70.51 | 70.52 | 70.30 |
| | MVCD | 1.86 | 2.59% | 0.987 | 14.28 | 2.11 | 12.17 | 69.71 | 70.19 | 60.60 | 71.57 | 71.70 | 69.00 |
| | ORE | 1.62 | 2.30% | 0.989 | 16.17 | 1.66 | 14.51 | 68.89 | 69.35 | 60.10 | 70.51 | 70.52 | 70.30 |
| | SemTopology | 0.69 | 0.98% | 0.995 | 11.81 | 0.43 | 11.38 | 69.82 | 70.22 | 62.30 | 70.51 | 70.52 | 70.30 |
| | OW-DETR | **0.30** | **0.43%** | **0.998** | 12.28 | 0.48 | **11.81** | 70.21 | 70.18 | 62.00 | 70.51 | 70.52 | 70.30 |
| | Ours | 0.87 | 1.24% | 0.994 | 29.03 | **-0.27** | 29.30 | 69.77 | 70.77 | 50.62 | 70.64 | 70.58 | 71.60 |

Table 15: Incremental learning results under the Four-Step scenario of VOC(5+5+5+5). The evaluation results of each subtask after each incremental learning step are presented in the table in a stepped presentation. Joint mAP, seen as the Upper Bound mAP, refers to the mAP of normal learning.

| Method | mAP | | | | mAP↑ | AbsGap↓ | RelGap↓ | $\Omega_{50}$ ↑ | Joint mAP |
|---|---|---|---|---|---|---|---|---|---|
| | A(1-5) | +B(6-10) | +B(11-15) | +B(16-20) | | | | | |
| SID | 70.60 | | | | 70.60 | 0.00 | 0.00% | 1 | 70.60 |
| | 49.60 | 68.00 | | | 58.80 | 11.30 | 16.12% | 0.919 | 70.10 |
| | 38.20 | 43.60 | 52.90 | | 44.90 | 27.20 | 37.73% | 0.821 | 72.10 |
| | 33.50 | 36.90 | 38.40 | 36.00 | 36.20 | 35.40 | 49.44% | 0.742 | 71.60 |
| ILOD | 66.30 | | | | 66.30 | 0.00 | 0.00% | 1 | 66.30 |
| | 42.90 | 61.10 | | | 52.00 | 13.80 | 20.97% | 0.895 | 65.80 |
| | 39.20 | 46.80 | 55.00 | | 47.00 | 23.50 | 33.33% | 0.819 | 70.50 |
| | 34.60 | 38.50 | 40.10 | 43.80 | 39.25 | 30.55 | 43.77% | 0.755 | 69.80 |
| RD-IOD | **71.97** | | | | **71.97** | 0.00 | 0.00% | 1 | **71.97** |
| | **66.23** | 69.98 | | | 68.10 | 20.03 | 22.73% | 0.886 | 88.13 |
| | **60.71** | 51.24 | 60.00 | | 57.32 | 16.85 | 22.72% | 0.849 | 74.17 |
| | **54.89** | 44.64 | 39.81 | 41.02 | 45.09 | 28.76 | 38.94% | 0.789 | 73.85 |
| CIFRCN | 63.90 | | | | 63.90 | 0.00 | 0.00% | 1 | 63.90 |
| | 43.80 | 71.20 | | | 57.50 | 13.68 | 19.22% | 0.904 | 71.18 |
| | 35.30 | 49.00 | 68.40 | | 50.90 | 22.48 | 30.64% | 0.834 | 73.38 |
| | 34.60 | 44.10 | 55.60 | 59.60 | 48.48 | 22.01 | 31.22% | 0.797 | 70.51 |
| ERD | 70.45 | | | | 70.45 | 0.00 | 0.00% | 1 | 70.45 |
| | 60.86 | 76.66 | | | 68.76 | 2.35 | 3.30% | 0.984 | 71.11 |
| | 48.33 | 65.51 | 73.63 | | 62.49 | 8.59 | 12.08% | 0.948 | 71.08 |
| | 41.25 | 57.38 | 63.57 | 53.12 | 53.83 | 16.77 | 23.57% | 0.902 | 70.60 |
| Ours | 70.34 | | | | 70.34 | 0.00 | 0.00% | 1 | 70.34 |
| | 62.17 | **77.00** | | | **69.86** | **0.50** | **0.71%** | **0.996** | 70.36 |
| | 50.50 | **69.66** | **74.72** | | **64.96** | **7.21** | **9.99%** | **0.964** | 72.17 |
| | 43.44 | **63.89** | **68.57** | **54.88** | **57.69** | **12.95** | **18.34%** | **0.927** | 70.64 |

Table 16: Incremental learning results under the Four-Step scenario of VOC(15+1+1+1+1+1).

| Scenarios | aero | bike | bird | boat | bottle | bus | car | cat | chair | cow | table | dog | horse | mbike | person | plant | sheep | sofa | train | tv | mAP↑ | SID | MVCD |
|---|---|---|---|---|---|---|---|---|---|---|---|---|---|---|---|---|---|---|---|---|---|---|---|
| A(1-20) | 74.4 | 75.8 | 73.1 | 54.5 | 60.6 | 72.8 | 81.3 | 84.4 | 56.3 | 70.4 | 58.2 | 81.2 | 79.1 | 77.9 | 82.5 | 50.9 | 69.1 | 62.3 | 76.3 | 71.6 | 70.6 | 71.6 | 71.6 |
| A(1-15) | 77.4 | 76.5 | 73.9 | 54.0 | 58.8 | 74.6 | 81.4 | 84.8 | 58.3 | 74.8 | 59.4 | 81.0 | 79.8 | 75.6 | 82.6 | | | | | | 72.9 | 72.6 | 73.7 |
| +B(16 plant) | 77.4 | 76.8 | 74.4 | 56.0 | 61.9 | 76.5 | 78.7 | 85.7 | 57.1 | 75.5 | 61.0 | 84.0 | 84.5 | 79.4 | 82.2 | 39.8 | | | | | **71.9** | 68.2 | 68.0 |
| +B(17 sheep) | 74.8 | 73.9 | 72.5 | 54.9 | 58.8 | 76.0 | 77.5 | 85.8 | 51.9 | 63.6 | 61.3 | 81.9 | 78.3 | 76.3 | 81.7 | 35.5 | 20.2 | | | | **66.2** | 65.6 | 63.0 |
| +B(18 sofa) | 74.1 | 68.2 | 68.4 | 54.7 | 55.0 | 75.1 | 77.3 | 85.3 | 49.0 | 61.0 | 60.7 | 79.6 | 77.6 | 75.9 | 77.7 | 31.4 | 17.7 | 51.1 | | | **63.3** | 63.3 | 57.3 |
| +B(19 train) | 68.1 | 67.6 | 67.1 | 50.5 | 52.0 | 68.1 | 77.9 | 84.7 | 44.3 | 56.0 | 60.9 | 79.8 | 71.5 | 74.6 | 77.1 | 29.5 | 10.4 | 22.8 | 60.6 | | **59.0** | 56.7 | 53.2 |
| +B(20 tv) | 63.8 | 66.5 | 66.2 | 48.7 | 52.6 | 64.3 | 77.1 | 85.0 | 44.8 | 46.3 | 58.2 | 75.5 | 71.8 | 74.9 | 77.0 | 30.1 | 6.3 | 21.0 | 54.0 | 45.5 | **56.5** | 51.9 | 48.9 |

Table 17: Main methods and base detectors for class-incremental object detection task in recent years.

| Method | Method Type | Base Detector |
|---|---|---|
| LwF (Li & Hoiem, 2018) | Pseudo-Labels | Faster-RCNN (Ren et al., 2015) |
| SID(Peng et al., 2021) | Knowledge Distillation | CenterNet(Duan et al., 2019) |
| ILOD (Shmelkov et al., 2017) | Knowledge Distillation | Faster-RCNN(Ren et al., 2015) |
| RILOD (Li et al., 2019) | Knowledge Distillation
External Data | RetinaNet(Lin et al., 2017) |
| Faster ILOD (Peng et al., 2020) | Knowledge Distillation | Faster-RCNN (Ren et al., 2015) |
| Meta-ILOD (Joseph et al., 2021b) | Knowledge Distillation
Meta-Learning
Exemplar Replay | Faster-RCNN(Ren et al., 2015) |
| RD-IOD (Yang et al., 2020) | Knowledge Distillation | Faster-RCNN(Ren et al., 2015) |
| CIFRCN (Hao et al., 2019) | Knowledge Distillation | Faster-RCNN(Ren et al., 2015) |
| MVCD (Yang et al., 2021a) | Knowledge Distillation | Faster-RCNN(Ren et al., 2015) |
| MMA (Cermelli et al., 2022) | Knowledge Distillation | Faster-RCNN(Ren et al., 2015) |
| DMC (Zhang et al., 2019) | Knowledge Distillation
External Data | RetinaNet(Lin et al., 2017) |
| ORE (Joseph et al., 2021a) | Pseudo-Labels
Exemplar Replay | Faster-RCNN(Ren et al., 2015) |
| SemTopology (Yang et al., 2021b) | Knowledge Distillation
Exemplar Replay | Faster-RCNN(Ren et al., 2015) |
| OW-DETR (Gupta et al., 2022) | Pseudo-Labels
Exemplar Replay | DETR(Carion et al., 2020) |
| IncDet (Liu et al., 2020) | Pseudo-Labels
EWC | Faster-RCNN(Ren et al., 2015) |
| ERD (Feng et al., 2022) | Knowledge Distillation | GFL v1(Li et al., 2020) |
| Ours | Knowledge Distillation | YOLOX(Ge et al., 2021) |

