# OpenReview forum: "Task Regularized Hybrid Knowledge Distillation For Incremental Object Detection"
_ICLR.cc/2024/Conference — Submitted to ICLR 2024_

### Official Review · Reviewer_71CT · 2023-10-31

**Soundness:** 2 fair
**Presentation:** 2 fair
**Contribution:** 2 fair
**Rating:** 5
**Confidence:** 5

**Summary:**

The paper works on the topic of incremental object detection. A soft knowledge distillation and a hard knowledge distillation are proposed to transfer the teacher's knowledge. A task regularization loss is further proposed to balance the training of old and new tasks. Experimental results are conducted on Pascal VOC and MS COCO.

**Strengths:**

(1) The paper is clearly written.
(2) The motivation of the paper is interesting to consider the confidence of each sample in the knowledge distillation process.
(3)  The results on standard benchmarks look good.

**Weaknesses:**

(1) The motivation is not well illustrated. In the abstract, knowledge fuzziness and imbalance learning are mentioned as the main causes of catastrophic forgetting, but it is unclear what it means in the learning process. It would be good to demonstrate the observation in the data and training process.
(2) The proposed methods do not seem very novel to me. It is a simple alternative to knowledge distillation and loss balance is also not new in the context of continual learning.
(3) In the experimental section, I notice that YOLOX is used for the proposed method, but it is not the same as other competing methods, is it fair to compare in this way?
(4) In the introduction section, it would be good to add some references when talking about the relevant methods.

**Questions:**

Please see the Weaknesses section. I think the novelty of the method and the fair comparison are critical for the rebuttal.

---

### Official Review · Reviewer_uijP · 2023-10-31

**Soundness:** 2 fair
**Presentation:** 2 fair
**Contribution:** 2 fair
**Rating:** 3
**Confidence:** 4

**Summary:**

This paper addresses the challenge of catastrophic forgetting in Incremental Object Detection (IOD). The authors propose a novel approach to tackle this problem by focusing on effective Knowledge Selection Strategy (KSS) and Knowledge Transfer Strategy (KTS).  The paper introduces an image-level hybrid knowledge representation method, HKR, which combines instance-level soft knowledge (logits) and hard knowledge (one-hot predictions) to strike a balance between knowledge reliability and ambiguity. A task regularized distillation method, TRD, is presented as an effective knowledge transfer strategy. Extensive experiments on various scenarios demonstrate that the proposed methods achieve state-of-the-art performance. Notably, the approach significantly reduces the mAP gap between incremental learning and joint learning, particularly in challenging scenarios.

**Strengths:**

1. The paper addresses the problem of catastrophic forgetting in Incremental Object Detection (IOD). While knowledge distillation is a known approach for overcoming catastrophic forgetting, this paper introduces innovative strategies for hybrid knowledge representation and task regularized distillation.
2. The empirical evaluation is conducted on multiple datasets, including MS COCO and Pascal VOC, showcasing the quality of the research. The results demonstrate the effectiveness of the proposed methods.

**Weaknesses:**

1. The proposed method may lack novelty as it extends existing knowledge distillation techniques with the hybrid knowledge representation strategy and task regularized distillation method, which are relatively common techniques in the field of continual learning.
2. For the HKR, extending classical logits distillation to include both logits and one-hot distillation, while potentially effective, might be considered an incremental extension of existing knowledge distillation techniques. In this sense, it may not be perceived as a highly novel contribution.
3. While the paper claims to focus on incremental object detection, the proposed strategies appear to be general and not specifically tailored to the challenges of object detection. There is a disconnect between the paper's focus and the proposed solutions.

**Questions:**

1.Could the authors provide more specific and tailored adaptations of their method for incremental object detection, given the paper's stated focus? How can the proposed strategies be extended to better address the challenges unique to object detection?
2. What aspects of the proposed method are particularly novel and unique compared to existing knowledge distillation techniques, especially those applied in the context of continual learning?
3.How does the proposed method compare to existing approaches for incremental object detection? Can the authors provide a more comprehensive discussion of the advantages and limitations compared to other methods in the field?
4.Many functions in this paper are redundant and general. For example, equations 4 and 5 are almost identical. Why not merge them?

---

### Official Review · Reviewer_b6i8 · 2023-11-01

**Soundness:** 2 fair
**Presentation:** 2 fair
**Contribution:** 2 fair
**Rating:** 3
**Confidence:** 4

**Summary:**

The paper proposes a technique for knowledge distillation for object detection. It proposes alternative loss functions based on hard and soft labels where the weights hard and soft labels are decided based on the difference between the top and second prediction. It performs a weighted average between new and old data to prevent catastrophic forgetting. Results are presented on the COCO dataset.

**Strengths:**

No major changes in existing codebase is needed to implement the paper, so its an easy thing to try.

**Weaknesses:**

Most of the design choices which are proposed are fairly well understood and have been adopted in prior literature in the form of weighted labels, psedo labels, semi-supervised settings etc, so the technical contribution is not very significant. We are using some pre-existing loss functions which have been used in similar computer vision problems and applying it to this setting. Experimental results on PASCAL are also missing, which is commonly used for this task.

**Questions:**

nothing specific

---

### Official Review · Reviewer_3fPp · 2023-11-05

**Soundness:** 3 good
**Presentation:** 3 good
**Contribution:** 2 fair
**Rating:** 3
**Confidence:** 4

**Summary:**

This paper aims to alleviate catastrophic forgetting in incremental object detection through knowledge distillation. It finds two reasons for catastrophic forgetting: knowledge fuzziness and imbalance learning, and further proposes combining hard and soft pseudo-labels for better knowledge selection. It also proposes a task-based regularization distillation loss to balance the learning process between new and old tasks. Extensive experiments demonstrate the effectiveness of the method.

**Strengths:**

1. Different from other methods that only use hard or soft pseudo-labels, this paper combines operations like mixup on hard and soft labels through adaptive thresholds, so that the student model can effectively learn the beneficial knowledge from the teacher.
2. The experiments are sufficient and demonstrate the effectiveness of this method on two popular benchmarks and under various experimental settings.

**Weaknesses:**

1. The novelty of this method is limited. The proposed distillation-based approach is a minor improvement over previous work, and its innovativeness does not meet the publication standards.
2. Is it unfair to compare results on different detectors? The method designed in this paper to alleviate catastrophic forgetting is not specifically designed for YOLOX. Why is YOLOX chosen as the basic framework? In order to make a fair comparison, it is recommended to show the results on the same detector, such as reimplementing other methods on YOLOX.
3. The description of Figure 1 is unclear, for example, whether w.o. and w/o are the same. There is no description of feature distillation in the article, but there are arrows in Figure 1, which brings a lot of confusion. And there is no description of NMS.
4. Eq. 7 and 8 are the same, without the effect of balancing the learning process of new and old tasks as claimed in the paper. The last term in Eq. 10 is a regular L2 loss. So the implementation of ” task regularized” distillation may require further explanation. In addition, can the author ablate the benefits brought by Loss_diff?

**Questions:**

Suggestions and questions for the authors are detailed in the Weaknesses.

---

### Meta-Review · Area_Chair_L9Jo · 2023-12-08

**Metareview:**

The paper consider incremental object detection, where catastrophic forgetting is mitigated through a knowledge distillation process by combining logits and one-hot predictions to trade-off soft knowledge and hard knowledge. The strength of the paper is the set of advantageous results demonstrated on standard benchmarks, while the weakness is the limited novelty of techniques.

The paper received four reviews, all of which deemed the design choices for knowledge distillation used here to be well-known. Further, concerns are raised on the experimental choices of using YOLOX for the proposed method while other methods use different backbones. No rebuttal was submitted. The AC agrees with the reviewer consensus that the paper may not be accepted.

**Justification For Why Not Higher Score:**

Limited technical contribution and sub-optimal empirical comparisons.

**Justification For Why Not Lower Score:**

Not applicable.

---

### Decision · Program_Chairs · 2024-01-16

Reject